# A Bayesian Approach to Evaluation of Soil Biogeochemical Models

Hua W. Xie[1], Adriana L. Romero-Olivares[2], Michele Guindani[3], and Steven D. Allison[4]

[1]Center for Complex Biological Systems, University of California, Irvine, 2620 Biological Sciences III Irvine, California 92697, United States of America

[2]Department of Natural Resources & the Environment, University of New Hampshire, 114 James Hall, Durham, New Hampshire 03824, United States of America

[3]Department of Statistics, University of California, Irvine, 2241 Donald Bren Hall, Irvine, California 92697, United States of America

[4]Department of Ecology and Evolutionary Biology, Department of Earth System Science, 321 Steinhaus Hall, University of California, Irvine, California 92697, United States of America

*Correspondence to*: Hua W. Xie (xiehw@uci.edu)

**Abstract.** To make predictions about the carbon cycling consequences of rising global surface temperatures, Earth system scientists rely on mathematical soil biogeochemical models (SBMs). However, it is not clear which models have better predictive accuracy, and a rigorous quantitative approach for comparing and validating the predictions has yet to be established. In this study, we present a Bayesian approach to SBM comparison that can be incorporated into a statistical model selection framework. We compared the fits of linear and non-linear SBMs to soil respiration data compiled in a recent meta-analysis of soil warming field experiments. Fit quality was quantified using Bayesian goodness-of-fit metrics, including the Widely Applicable information criterion (WAIC) and Leave-one-out cross-validation (LOO). We found that the linear model generally out-performed the non-linear model at fitting the meta-analysis data set. Both WAIC and LOO computed higher overfitting risk and effective numbers of parameters for the non-linear model compared to the linear model, conditional on the data set. Goodness-of-fit for both models generally improved when they were initialized with lower and more realistic steady state soil organic carbon densities. Still, testing whether linear models offer definitively superior predictive performance over non-linear models on a global scale will require comparisons with additional site-specific data sets of suitable size and dimensionality. Such comparisons can build upon the approach defined in this study to make more rigorous statistical determinations about model accuracy while leveraging emerging data sets, such as those from long-term ecological research experiments.

## 1 Introduction

Coupled Earth system models (ESMs) and constituent soil biogeochemical models (SBMs) are used to simulate global soil organic carbon (SOC) dynamics and storage. As global climate changes, some ESM and SBM simulations suggest that substantial SOC losses could occur, resulting in greater soil $CO_2$ emissions (Crowther et al., 2016). However, there is vast divergence between model predictions. For instance, one ESM predicts a global SOC loss of 72 Pg C over the 21[st] century, while another predicts a gain of 253 Pg C (Todd-Brown et al., 2014).

Soil biogeochemical models vary greatly in structure (Manzoni and Porporato, 2009), but can be broadly partitioned into two categories: those that implicitly represent soil C dynamics as first-order linear decay processes and those that explicitly represent microbial control over C dynamics with non-linear Michaelis-Menten functions (Wieder et al., 2015a). Explicit models typically include more parameters than linear models because multiple microbial parameters are needed for each decay process as opposed to a single rate parameter. The additional parameters allow explicit models to represent microbial mechanisms, but at the expense of greater model complexity.

Rigorous statistical approaches should be applied to investigate how explicit representation of microbial processes affects predictive model performance. ESM and SBM comparisons involving empirical soil C data assimilations have been conducted previously (Allison et al., 2010; Li et al., 2014) but few standardized statistical methods for ESM and SBM benchmarking and comparison have been developed that would allow for rigorous model selection. Prior model comparisons have involved graphical qualitative comparisons or use of basic fit metrics such as the coefficient of determination, $R^2$, to judge fit quality. However, these simple approaches are

| 48 | insufficient for comparing an increasing number of complex models (Jiang et al., 2015; Luo et al., 2016; Wieder et al., 2015a). |
| 50 | $R^2$ on its own provides limited information about goodness-of-fit. In unmodified form, it quantifies the extent to which the variation of just one chosen model outcome—for instance the mean outcome for a range of |
| 52 | parameter values—corresponds to the variation in the data set (Gelman et al., 2019). $R^2$ does not capture model complexity, overfitting, or parameter uncertainty, which is a reason why $R^2$ by itself is not sufficient for model |
| 54 | evaluation (Kvålseth, 1985). Without accounting for model complexity and parameter count, focusing on optimizing fit by $R^2$ values alone can easily lead to overfitting (Spiess and Neumeyer, 2010). |
| 56 | Encouragingly, a rich toolset to further inform quantitative model evaluation and comparison can be drawn from Bayesian statistics (Hararuk et al., 2014, 2018; Hararuk and Luo, 2014). These tools include information |
| 58 | criteria and approximate cross-validation, goodness-of-fit metrics designed for the simultaneous comparison of multiple structurally diverse models. Like $R^2$, information criteria and cross-validation are quantitative measures that |
| 60 | estimate the fit quality of a model to a given data set. Differing from $R^2$, information criteria and cross-validation are relative rather than absolute measures. These metrics evaluate the extent to which the data set supports particular |
| 62 | distributions of parameter values and in turn, the uncertainty of parameter estimates. Consequently, if the distribution of Model A outcomes aligns more closely to the data set than the distribution of Model B outcomes, we |
| 64 | regard Model A as being more likely to explain the data compared to Model B. Information criteria and cross-validation metrics also typically include terms penalizing for model complexity and overfitting as part of their |
| 66 | computation (Gelman et al., 2014). Hence, information criteria and approximate cross-validation are useful tools for model evaluation because they present a comprehensive summary of model fit to time series data and can estimate |
| 68 | model predictive accuracy for unmeasured and out-of-sample data points. |
| 70 | Examples of information criteria popularized by widely used R packages such as lme4 and rjags include the Akaike information criterion (AIC), Bayesian information criterion (BIC), and deviance information criterion (DIC) |
| 72 | (Vehtari and Ojanen, 2012). However, these metrics have some limitations. AIC, BIC, and DIC do not use full sampled posterior distributions in their computational processes. AIC and BIC both rely on a pointwise maximum |
| 74 | likelihood estimate that cannot be derived from non-uniform Bayesian prior distributions, including normal distributions. AIC and BIC (despite BIC's name) thereby have limited use in Bayesian statistics settings. DIC can |
| 76 | accommodate non-uniform priors but is calculated from pointwise simplified posterior means. The compression of full posteriors into pointwise means can prompt DIC to compute an impossible negative effective model parameter |
| 78 | count in select situations (Gelman et al., 2014). Consequently, the original forms of AIC, BIC, and DIC are no longer recommended for use in Bayesian model assessment by some statisticians in light of superseding alternatives |
| 80 | (Gelman et al., 2014). |
| | Three predictive goodness-of-fit metrics address the limitations and stability issues of AIC, BIC, and DIC by incorporating full, non-uniform posterior distributions in their calculations to better account for overfitting and |
| 82 | model size (Christensen et al., 2010; Gelman et al., 2014). These metrics include the Widely Applicable information criterion (WAIC), log pseudomarginal likelihood (LPML), and Pareto-smoothed important sampling leave-one-out |
| 84 | cross-validation (PSIS-LOO and hereby referred to as LOO). WAIC, LPML, and LOO can estimate the ability of models to fit unobserved measurements outside of the set of measured data samples (Vehtari et al., 2017). Thus, |
| 86 | WAIC, LPML, and LOO can be considered as superior barometers for model predictive accuracy compared to AIC, BIC, and DIC. |
| 88 | The overarching goal of this study was to develop a statistically rigorous and mathematically consistent data assimilation framework for SBM comparison that uses predictive Bayesian goodness-of-fit metrics. We |
| 90 | pursued three specific objectives as part of that goal. First, we compared the behaviors of two different SBMs, a linear microbial-implicit model termed the conventional model (CON) and a non-linear microbial-explicit model |
| 92 | called the Allison-Wallenstein-Bradford model (AWB) (Fig. 1), following data assimilation with soil respiration data sourced from a meta-analysis of soil warming studies (Romero-Olivares et al., 2017). Second, we characterized |
| 94 | the parameter spaces of these models using prior probability distributions of parameter values informed by previous studies and expert judgment. Third, we compared specific Bayesian predictive information criteria in WAIC, LPML, |
| 96 | and LOO, to the coefficient of determination, $R^2$, for quantifying goodness-of-fit to data. AIC, BIC, and DIC were not analyzed due to their stability limitations, our usage of non-uniform prior distributions, and redundancy with |
| 98 | WAIC. |

## 2 Methods

## 2.1 Model Structures

We compared two SBMs, the CON and AWB models (Allison et al., 2010). The models were selected for this study due to their relative equation simplicity, their tractable parameter count, and limited biological data input requirements (Supplemental Index 1). The CON system models three separate C pools as state variables including
SOC, dissolved organic C (DOC), and microbial biomass C (MIC) pools, while AWB includes SOC, DOC, MIC, and extracellular enzyme biomass C (ENZ) pools (Fig. 1). Additionally, these models were chosen because they are
C-only models without nitrogen (N) pools. The increased complexity of N-accounting SBMs will require future studies with coupled N data sets (Manzoni and Porporato, 2009).
**2.2 Meta-analysis Data**

The data set for model fitting was compiled from a recent meta-analysis of 27 soil warming studies that

measured $CO_2$ fluxes (Romero-Olivares et al., 2017). The experiments reported between 1 and 13 years of $CO_2$ flux measurements following warming perturbation. The elements of this data set consisted of empirical response ratios
calculated by dividing $CO_2$ fluxes measured in the warming treatments by time-paired $CO_2$ fluxes measured in the control treatments. We calculated an annual mean response ratio for each experiment (if data were available for that
year) after warming treatment began. Using these annual means, we calculated one overall mean response ratio for each year along with pooled variances and standard deviations. Pooled data points were assumed to be "collected" at
the halfway point of each year. Because the experiments had variable lengths, the sample size for the pooled annual mean declines with increasing time since warming perturbation. The warming perturbation was 3°C on average
across all the studies, and this average was used as the magnitude of warming in the model simulations.

Model-outputted response ratios were calculated by dividing simulated $CO_2$ flux following warming

perturbation by the $CO_2$ flux at pre-warming steady state. We fit models to flux response ratios rather than raw flux measurements for several reasons (Wieder et al., 2015b). First, we eliminate the need to convert flux measurements
from different experiments into a common unit. Second, response ratios represent a standardized metric for warming response across disparate ecosystem types with varying climate, soil, and vegetation properties. Finally, fitting a
mean response ratio overcomes data gaps present in individual experiments.

**2.3 Hamiltonian Monte Carlo Fitting of Differential Equation Models**

CON and AWB ordinary differential equation systems were simulated using the CVODE backward differentiation method (Curtiss and Hirschfelder, 1952) from the SUNDIALS library of equation solvers
(Hindmarsh et al., 2005). Differential equation models contain parameters that affect state variables, and model-fitting through Markov chain algorithms involves iterating through parameter space one set of parameters at a time.
We performed model fitting using a Markov chain algorithm called the Hamiltonian Monte Carlo (HMC), using version 2.18.1 of the RStan interface to the Stan statistical software (Carpenter et al., 2017; Guo et al., 2019) and
version 3.4.1 of R (R Core Team, 2017). HMC is not a random walk algorithm and uses Hamiltonian mechanics to determine exploration steps in parameter space. HMC has been theorized to offer more efficient exploration of high-
dimensional parameter space than traditional Random-Walk Metropolis algorithms (Beskos et al., 2013).

Conditional on the meta-analysis data set, the HMC algorithm computed posterior and posterior predictive

distributions, from which Bayesian statistical inferences on likely ranges of parameter values were then made. Posterior distributions are the distributions of more likely model parameter values conditional on the data. Posterior
predictive distributions are the distributions of more likely values for unobserved data points from the data-generating process conditional on the observations. In the case of this study, the experiments constituting the meta-
analysis would be the data-generating process.

For the sake of clarity, it is important to distinguish between the frequentist confidence intervals and

Bayesian posterior predictive intervals and distributions we describe in our study. Confidence intervals are calculated from the sample means and standard errors at observed data points and indicate ranges of values that are
likely to contain the true data values with repeated sample collections using the same methodology. Posterior predictive intervals and distributions are computed after estimation of the posterior parameter distributions and
represent the likely distributions of unobserved data values conditional on observed data values. Bayesian credible intervals, which we will also discuss in this study, are ranges of values that parameters are likely to take with some
probability that are conditional on the observed data. Credible areas indicate the probability densities of parameter values across credible intervals.
We ran four chains for 35,000 iterations each for our HMC simulations, with the first 10,000 iterations being discarded as burn-in for each chain. Hence, our posterior distributions consisted of 100,000 posterior samples
per HMC run. In retrospect, because our credible areas displayed sufficient smoothness (Supplemental Fig. 2) and

Bayesian diagnostics indicated adequate posterior sampling (Supplemental Table 5), we could have reduced
simulation time without impairing posterior computation by running shorter chains that consisted of 20,000 to
30,000 iterations. To minimize the presence of divergent energy transitions, which indicate issues with exploring the
geometry of the parameter space specified by the prior distributions, we set the adaptation delta to 0.95, the initial
step size to 0.1, and maximum tree depth to 12. Those parameters determine how the HMC algorithm proposes new
sets of parameters at each step and were set so that the HMC would begin with smaller exploration steps. The
algorithm varies the step size from its initial value throughout posterior sampling to maintain a desired acceptance
rate; the tuning sensitivity of the step size is governed by the adaptation delta value, with higher values indicating
reduced sensitivity.
We further constrained our HMC runs to characterize parameter regimes corresponding to higher biological
realism. Normal informative priors were used to initiate the runs, and the prior distribution parameters were chosen
based on expert opinion and previous empirical observations (Allison et al., 2010; Li et al., 2014). Prior distributions
had non-infinite supports; supports were truncated to prevent the HMC from exploring parameter space that was
unrealistic (Supplemental Table 2).

### 2.4 Model Steady State Initialization

Because we were mainly interested in testing model predictions of soil warming response, the models were
initiated at steady state prior to the introduction of warming perturbation to isolate model warming responses from
steady state attraction. We fixed pre-perturbation steady state soil C densities to prevent HMC runs from exploring
parameter regimes corresponding to biologically unrealistic C pool densities and mass ratios.
To set pre-warming steady state soil C densities, we first analytically derived steady state solutions of the
ordinary differential equations of the models. Then, with the assistance of Mathematica version 12, we re-arranged
the equations by moving the steady state pool sizes to the left-hand side (Supplemental Appendix 2), such that we
could determine the value of parameters dependent on pool sizes while allowing the rest of the parameters to vary
for the HMC. Consequently, we could constrain the pre-warming pool sizes from reaching unrealistic values in the
simulations.

### 2.5 Sensitivity Analysis of C Pool Ratios

Sensitivity analyses examine how the distributions of model input values influence the distributions of
model outputs. In our study, we considered pre-warming C-pool densities as a model input. We performed a
sensitivity analysis to observe how the choice of pre-warming C pool densities and C-pool ratios would affect the
model fits and posterior predictive distribution of C pool ratios.
      We compared the model outputs and post-warming response behavior of AWB and CON at equivalent C
pool densities and ratios. The ratio of soil microbe biomass C (MIC) density to SOC density has been observed to
vary approximately from 0.01 to 0.04 (Anderson and Domsch, 1989; Sparling, 1992), so we used those numbers as
guidelines for establishing the ranges of the C pool densities and density ratios explored in our simulations. One
portion of the analysis involved running HMC simulations in which we set the pre-warming MIC density at 2 mg C
$g^{-1}$ soil and then varied the SOC density from 50 to 200 mg C $g^{-1}$ soil in increments of 25, stepping from 0.04 to 0.01
with respect to the MIC-to-SOC ratio.  A second portion of the analysis involved observing the effect of varying
pre-warming MIC from 1 to 8 mg C $g^{-1}$ soil while holding pre-warming SOC at 100 mg C $g^{-1}$ soil.
      For some combinations of the prior distributions and pre-warming steady state C pool densities
(Supplemental Table 2), AWB HMC runs wandered into unstable parameter regimes that would prevent the
algorithm from reliably running to completion. Consequently, we do not compare simulation results for AWB and
CON with pre-warming SOC densities below 50 mg C $g^{-1}$ soil. Other combinations of prior distribution and pre-
warming C pool density choices that were not necessarily biologically realistic allowed stable AWB runs with lower
pre-warming SOC densities.

### 2.6 Information Criteria and Cross-validation

In addition to $R^2$, we used the WAIC, LPML, and LOO Bayesian predictive goodness-of-fit metrics to
evaluate models with the meta-analysis warming response data. LPML is an example of cross validation that is
calculated similarly to LOO (Gelfand et al., 1992; Gelfand and Dey, 1994; Ibrahim et al., 2001) but differs from
LOO in how the importance ratio sampling portion of its computation is handled. For further explanation regarding
importance ratios and their role in evaluating approximate cross-validation metrics, refer to the description of the
LOO algorithm presented in Vehtari and Ojanen (2012). LOO updates LPML by implementing a smoothing process in which the largest importance ratios are fitted with a Pareto distribution and then replaced by expected values from the distribution, which stabilizes the importance ratio sampling.
Algorithmic differences between WAIC and LPML and LOO render them appropriate for different statistical modeling goals and make them complementary metrics. WAIC is suitable for estimating the relative
quality of model fits to hypothetical repeated samples collected at existing experimental time points, whereas LOO and LPML are suitable for estimating the quality of fits to hypothetical measurements taken between observed time
points (Vehtari et al., 2017).

We used version 2.0.0 of the loo package available for R to calculate our WAIC and LOO values (Vehtari

et al., 2019). A lower WAIC and LOO and a higher LPML indicate a more likely model for a given data set. LPML can be multiplied by a factor of -2 to occupy a similar scale to LOO.
**3 Results**

**3.1 Parameter Posterior Distributions**

We obtained distributions of posterior predictive fits to the univariate response ratio data for both AWB and CON across different pre-warming MIC-to-SOC ratios. Posterior samples totaled 100,000 for each simulation.
Sampler diagnostics for the HMC runs indicated that the statistical models were valid at all pre-warming steady state values observed (Supplemental Table 6), that model parameter values converged across the four Markov chains
(Supplemental Fig. 7), and that the posterior parameter space was effectively sampled and explored (Supplemental Fig. 5) to generate enough independent posterior samples for inference (Supplemental Fig. 6). The ratios of effective
posterior parameter samples to total samples for parameters were generally satisfactory; across observed MIC-to-SOC ratios, they were all greater than 0.25 and mostly greater than 0.5 (Supplemental Table 5).
We also tracked divergent transitions, which mark points in chains at which the HMC algorithm was inhibited in its exploration and posterior sampling, potentially due to the parameter space becoming geometrically
confined and difficult to navigate. Divergent transitions occurred in the AWB HMC runs (Supplemental Fig. 9), though the ratios of divergent transitions to sampled iterations was relatively low for all runs. The highest divergent
transition ratio observed was 0.0217, corresponding to the simulation initiated with pre-warming SOC = 200 mg C $g^{-1}$ soil. There were no divergent transitions in the CON runs.
**3.2 Model Behaviors**

The CON curve monotonically decreases in response ratio over time, whereas the AWB curve displays

changes in slope sign (Fig. 2). The difference in curve shape (Fig. 3a, b) is in line with CON's linear status and AWB's non-linear formulation with more parameters (Allison et al., 2010). By 50 years after warming, mean fit
curves for AWB and CON return to 1.0 after their initial increase (Fig. 3c, d), consistent with prior observations and expectations at steady state (van Gestel et al., 2018; Romero-Olivares et al., 2017).
From a cursory visual evaluation, neither of the models clearly out-performs the other across all prewarming steady states. The 95% confidence interval of the first data point at t = 0.5 years does not include the
AWB SOC100 posterior predictive mean as it does for the CON SOC100 mean (Fig. 2), which most likely impaired AWB's quantitative goodness-of-fit metrics. However, the 95% response ratio posterior predictive interval suggests
that AWB is able to replicate the response ratio increase in the data from 1.5 to 3.5 years following the warming perturbation, which CON does not. The shape of the AWB posterior predictive interval also fits the data points and
confidence intervals occurring eight years or more after the perturbation more closely than that of CON (Fig. 3a, b).

For both AWB and CON, increasing the pre-warming SOC to higher densities from SOC = 50 to 200 mg C

$g^{-1}$ soil (hereby labeled from SOC50 to SOC200) while holding pre-warming MIC at 2 mg C $g^{-1}$ soil, DOC at 0.2 mg C $g^{-1}$ soil, and ENZ at 0.1 mg C $g^{-1}$ soil, corresponded to lower initial mean response ratios in the first year at the t =
0.5 year time point, which certainly inhibited the quantitative goodness-of-fit (Fig. 3a, b). For CON, increasing pre-warming SOC also reduced the magnitude of the mean fit slope. For AWB, increasing pre-warming SOC had no
clear effect on the curve slope, but the model needed more time to achieve peak mean response ratio from a lower start, with the peak being reached at t = 1.5 years in the SOC50 case and t = 3.5 years in the SOC200 case (Fig. 3b).
At higher pre-warming SOC, CON's reduced slope magnitude and AWB's lagging response ratio peak caused both models to exhibit slower returns to the steady state response ratio of 1.0 (Fig. 3c, d). On their trajectories back to
steady state, the mean SOC200 CON curve substantially overshoots the data means after t = 7.5 years (Fig. 3a),

whereas the SOC200 AWB curve exceeds the data means at a more moderate extent through the t = 8.5, 9.5, 10.5 and 11.5 year time points (Fig. 3b).

Changing the pre-warming MIC-to-SOC steady state pool size ratio by increasing pre-warming MIC from 1 to 8 mg C g$^{-1}$ soil (hereby labeled from MIC1 to MIC8) while holding pre-warming SOC at 100 mg C g$^{-1}$ soil had marginal to moderate qualitative effects on the mean response ratio curves for CON and AWB. The CON MIC1 and MIC8 curves are visually indistinguishable (Supplemental Fig. 1a, b), while the AWB MIC1 and MIC8 curves differ with the MIC8 curve displaying more gradual changes in slope and lower slope magnitudes (Supplemental Fig. 1c, d).

**3.3 Sensitivity Analysis of Parameter Distributions to Pre-warming C Pool Densities and Density Ratios**

In addition to response ratio fits, we observed the influence of pre-warming MIC-to-SOC ratios on model SOC stock response ratios in AWB and CON simulations following warming. Similar to the model flux response ratios, SOC response ratios were calculated by dividing evolved post-warming SOC densities by pre-warming densities. The SOC response ratios at 12.5 years for CON and AWB increased as pre-warming SOC was raised (and hence, the MIC-to-SOC ratio decreased) with other pre-warming C densities held constant, indicating reduced proportional SOC loss when SOC stocks were initiated at higher pre-warming densities (Supplemental Fig. 3a). For CON, SOC loss decreased from 27.1% at SOC50 to 9.2% at SOC200. In a similar trend for AWB, SOC loss decreased from 17.2% at SOC50 to 8.1% at SOC200. In contrast, raising pre-warming MIC densities (and hence, increasing the MIC-to-SOC ratio) with other pre-warming C densities held constant did not produce a shared trend for CON and AWB (Supplemental Fig. 3b). CON SOC loss decreased from 18.8% at MIC1 to 17.4% at MIC8, while AWB SOC loss increased from 11.3% at MIC1 to 16.3% at MIC8.

Truncation of prior supports, or distribution domains, generally did not prevent posterior densities from retaining normal distribution shapes. Deformation away from Gaussian shapes for the densities of $Ea_S$ from CON was observed at SOC50 and SOC75. For AWB, deformation was observed for the densities of $Ea_V$, $Ea_K$, and $E_{C_{ref}}$. All CON and AWB parameter posterior densities were otherwise observed to be Gaussian from SOC100 to SOC200. Example posterior densities and means for select model parameters at pre-warming SOC100 are presented in Fig. 4 and Supplemental Fig. 2. Parameter posterior means corresponding to other pre-warming C pool densities and ratios are presented in Supplemental Table 3.

**3.4 Sensitivity Analysis of Quantitative Fit Metrics to Pre-warming C Pool Densities and Density Ratios**

For both CON and AWB, LOO, WAIC, LPML, and R$^2$ all worsened as pre-warming steady state SOC density was increased from SOC50 to the less biologically realistic SOC200 (Fig. 5). CON's LOO and WAIC values increased respectively from -15.704 and -15.818 at SOC50 to -6.891 and -6.966 at SOC200, while AWB's LOO and WAIC values increased respectively from -11.028 and -11.379 at SOC50 to -5.97 and -6.579 at SOC200 (Supplemental Table 4a, b). Compared to AWB's metrics, CON's goodness-of-fit metrics deteriorated at a faster rate with the increase of pre-warming SOC. Nonetheless, CON outperformed AWB in LOO, WAIC, and LPML across all observed pre-warming SOC densities. The Bayesian metrics accounted for AWB's larger model size and increased propensity for overfitting as demonstrated by the consistently higher effective parameter counts associated with AWB (Supplemental Fig. 8a, b).

Varying pre-warming steady state MIC from MIC1 to MIC8 modestly impaired goodness-of-fit across the various metrics (Supplemental Fig. 4). CON's LOO and WAIC values increased respectively from -11.963 and -12.035 at MIC1 to -11.731 and -11.802 at MIC8, while AWB's LOO and WAIC values increased respectively from -8.63 and -9.302 at MIC1 to -8.181 and -8.711 at MIC8 (Supplemental Table 4c, d). CON did not deteriorate in goodness-of-fit at a faster rate than AWB with respect to increasing pre-warming MIC. Increasing pre-warming MIC has the opposite effect on MIC-to-SOC ratio compared to increasing pre-warming SOC, but both changes worsened goodness-of-fit across all metrics, indicating that changes to pre-warming MIC-to-SOC ratio did not produce consistent trends.

**4 Discussion**

Our study develops a quantitative, data-driven framework for model comparison that could be applied across different research questions, ecosystems, and scales. We demonstrated the novel deployment of WAIC and LOO, two more recently developed Bayesian goodness-of-fit metrics that estimate model predictive accuracy, to

evaluate SBMs using data from longitudinal soil warming experiments. WAIC and LOO improve upon older and more frequently used metrics, such as AIC and DIC, by accounting for model complexity and overfitting of data in a
more comprehensive, stable, and accurate fashion. The quantitative agreement between WAIC, LOO, and LPML reinforces the reliability and validity of information criteria and cross-validation metrics to complement use of
frequentist $R^2$.

We constrained the fitting of AWB and CON to biologically reasonable parameter space by fixing pre-
warming steady state C pool densities and establishing prior distributions informed by expert judgment (Supplemental Table 2). We observed that, despite the qualitative difference in the shapes of their mean posterior
predictive fit curves, CON and AWB could both potentially account for the soil warming response in the meta-analysis data set. For both models, posterior predictive fit distributions overlapped with the confidence intervals of
the data points (Fig. 2). However, with respect to the Bayesian goodness-of-fit metrics, CON quantitatively outperformed AWB across all pre-warming SOC and MIC densities observed (Fig. 5 and Supplemental Fig. 4)
because the Bayesian metrics adjusted for AWB's larger model size and consistently higher effective parameter count (Supplemental Fig. 8). For both models, lower pre-warming SOC densities corresponded to better warming
response fits (Fig. 5).

## 4.1 Model Responses to Warming over Time

After fitting, the response ratio curves of CON and AWB both trended toward the pre-warming steady state response ratio of 1.0 following the soil warming perturbation (Fig. 3). The settling of the curves to the pre-warming
model steady states aligns with previous literature which demonstrated that the magnitude of $CO_2$ flux tends to fall after reaching a post-warming maximum (Crowther et al., 2016; Romero-Olivares et al., 2017). In the meta-analysis
data set, this peak is reached immediately at the first data point at t = 0.5 years (Fig. 2). CON matched this data pattern in all of our observed simulations in outputting maximum response ratios at the first time point after
warming (Fig. 3a, c and Supplemental Fig. 1a, b). AWB was unable to output maximum response ratios at the first time point (Fig. 3b, d) and was therefore penalized in quantitative goodness-of-fit. Examining AWB's system of
equations (Supplemental Appendix 1b), we surmise that one reason for the later peak was due to the slower growth of MIC in the biologically truncated parameter space that AWB was limited to. MIC is a driving force for the
increase of $CO_2$ as a numerator term in the AWB flux equation (Supplemental Appendix 1b, Equation A10). Unlike MIC biomass in CON (Supplemental Appendix 1a, Equation A3), MIC biomass growth in AWB has two
loss terms in its differential equation (Supplemental Appendix 1b, Equation A8).

This is not to say that CON was clearly superior from a qualitative standpoint. CON's mean posterior
predictive curves were not able to match a subsequent local data maximum in the meta-analysis data set at t = 3.5 years, a trend which AWB's curves were able to replicate. The mean CON curves also substantially overshoot the
data at later time points following t = 7.5 years (Fig. 2a, Fig. 3a, c, and Supplemental Fig. 1a, b) because of the inability of first order linear models such as CON to display oscillatory dynamics (Hale and LaSalle, 1963).
In contrast, AWB displays damped oscillations in its response ratios following warming due to its non-linear dynamics (Fig. 2 and Fig. 3). AWB was able to match the points after t = 7.5 years more closely than CON.
The presence of respiration oscillations has been observed in long-term warming experiments, such as the one taking place at Harvard Forest (Melillo et al., 2017). It is possible AWB would be quantitatively rewarded in goodness-of-
fit metrics over CON for its ability to replicate biologically realistic oscillations in larger, site-specific data sets such as those from Harvard Forest.

## 4.2 Sensitivity Analyses of C Pool Densities and Density Ratios


We performed a goodness-of-fit sensitivity analysis to check whether the response ratio trends stayed
consistent, biologically realistic, and interpretable across a range of pre-warming, steady state soil C densities and pool-to-pool density ratios. For instance, we imposed constraints to reflect that MIC-to-SOC density ratios range
between 0.01 and 0.04 across various soil types (Anderson and Domsch, 1989; Sparling, 1992). CON and AWB response ratio curves exhibited realistic values and qualitatively consistent shapes across all pre-warming SOC and
MIC steady state densities, even at less realistic SOC densities above 100 mg C g$^{-1}$ soil (Fig. 3). There was enough uncertainty in the data that the 95% posterior predictive intervals for the model output always overlapped with the
95% confidence intervals of each fitted data point (Fig. 2). In most cases, the posterior mean response ratio curve also fell within the 95% data confidence interval.
We were unable to initiate our pre-warming SOC steady state density below SOC50 with the priors and MIC-to-SOC ratios used for AWB. Under SOC50, AWB HMC runs would not reliably run to conclusion and would

terminate due to ODE instabilities. Even at SOC50, we saw a reduction in independent and effective samples for certain parameters, namely $Ea_K$ and $E_{C_{ref}}$ (Supplementary Table 5). We did not drop under SOC50 for CON, as we sought to compare AWB and CON at similar MIC-to-SOC ranges. Our experience underscores the challenge of choosing realistic steady state soil C densities, density ratios, and prior distributions to obtain valid model comparisons limited to biologically realistic regimes.

The information criteria and cross-validation fit metrics generally indicated higher relative probability and predictive performance at lower pre-warming SOC values for AWB and CON (Fig. 5). The fit results suggest that SOC density of the soil at the sites included in the meta-analysis was likely closer to the lower end of the SOC density ranges examined in our sensitivity analysis. A less pronounced trend toward better fits was observed as pre-warming MIC density was decreased while pre-warming SOC density was held constant (Supplemental Fig. 4). No clear relationship was observed between MIC-to-SOC ratio and goodness-of-fit in the AWB and CON models.

The worsening IC and CV results at higher SOC densities support the notion that pre-warming steady state SOC densities should not be initialized over SOC100 in AWB and CON when fitting to this meta-analysis data set. Pre-warming SOC density was not observed to exceed 50 mg SOC g$^{-1}$ soil at sites included in the meta-analysis, reaching a maximum of 45 mg SOC g$^{-1}$ soil for the top 20 cm in one study with alpine wetland soil (Zhang et al., 2014). The majority of the $CO_2$ respired by soil microbes is sourced from surface soil (Fang and Moncrieff, 2005), and it is well-documented that SOC densities increase toward the soil surface (Jobbágy and Jackson, 2000). $^{14}$C measurements of $CO_2$ fluxes suggest that SOC densities representing the source of most heterotrophic respiration range between 40 to 80 mg SOC g$^{-1}$ soil (Trumbore, 2000), so the effective SOC densities associated with soil respiration at some meta-analysis sites may have been in this range.

Overall, the Bayesian metrics from the goodness-of-fit sensitivity analysis suggest that CON is superior to AWB at explaining the meta-analysis data set when accounting for model parsimony, particularly when the models are initiated in more realistic ranges of pre-warming SOC densities under SOC100. However, we caution against using these results to conclude that CON is a comprehensively superior predictive model over AWB without comparisons involving other longitudinal soil warming data sets. And other data aside, we observe that AWB has a useful advantage over CON conditional on the meta-analysis data set alone: AWB was more tolerant of changes in pre-warming conditions, displaying less IC and CV than CON as pre-warming SOC is increased (Fig. 5a – c). AWB's compensatory ability stemming from its larger model size could be more quantitatively rewarding in goodness-of-fit sensitivity analyses conducted on data assimilations with larger data sets.

For an additional check on the biological realism and plausibility of our simulations, we conducted a sensitivity analysis examining changes in model SOC stocks following warming. The response ratios of post-warming SOC stocks after 12.5 years, evaluated as the ratio of post-warming to pre-warming SOC densities, was computed from observed CON and AWB simulations at the posterior parameter means. SOC losses indicated by the response ratios ranged from 8.13 to 27.1% across both models (Supplemental Fig. 3). These results aligned with a recent comprehensive meta-analysis of 143 soil warming studies (Supplemental Fig. 10). The largest loss of 27.1%, occurring in CON at SOC50, is sizable, but the meta-analysis included 7 studies measuring losses greater than 20%, with the maximum loss observed at 54.4% (van Gestel et al., 2018).

Raising pre-warming SOC reduced SOC loss after 12.5 years of warming for both models (Supplemental Fig. 3a). For CON, SOC loss decreased from 27.1% at SOC50 to 9.2% at SOC200. For AWB, SOC loss decreased from 17.2% at SOC50 to 8.13% at SOC200. Varying pre-warming MIC affected the SOC response ratio more substantially for AWB than CON (Supplemental Fig. 3b). For AWB, SOC loss increased from 11.4% at MIC1 to 16.3% at MIC8, while SOC loss decreased from 18.8% at MIC1 to 17.4% at MIC8 for CON. The larger effect of increasing MIC on the SOC response ratio in AWB is likely due to MIC influence on SOC-to-DOC turnover, which is not a feedback accounted for in the equations of the CON model (Supplemental Appendix 1a).

The posterior means for the Arrhenius activation energy parameters $Ea$ of CON and AWB returned by the HMC simulations across the observed pre-warming C densities (Supplemental Table 3) differed somewhat from the parameter values used in Allison et al. (2010) and Li et al. (2014), which were in turn tuned based on activation energies estimated in a prior empirical analysis of enzyme-catalyzed soil organic matter decomposition processes (Trasar-Cepeda et al., 2007). In Allison et al. (2010), CON parameters $Ea_S$, $Ea_D$, and $Ea_M$ were respectively set at 47, 40, and 40 kJ mol$^{-1}$ and AWB parameters $Ea_V$ and $Ea_{VU}$ were both set at 47 kJ mol$^{-1}$. The AWB Michaelis-Menten $K_M$ terms were not parameterized to have Arrhenius temperature dependence in Allison et al. (2010). In Li et al. (2014), CON parameters $Ea_S$, $Ea_D$, and $Ea_M$ were set at 47, 47, and 20 kJ mol$^{-1}$ and AWB parameters $Ea_V$, $Ea_{VU}$, $Ea_K$, and $Ea_{KU}$ were set at 47, 47, 30, and 30 kJ mol$^{-1}$. These values were in line with the activation energies calculated in Trasar-Cepeda et al. (2007), which ranged from 17.0 to 57.7 kJ mol$^{-1}$, with the energies corresponding

to the decomposition of plant litter and protected organic matter being on the higher end and the energies corresponding to microbial biomass degradation being on the lower.

Our HMC simulations arrived at higher $Ea$ values, with the posterior means of $Ea_S$, $Ea_D$, and $Ea_M$ respectively ranging from 51.3 to 77.6 kJ mol$^{-1}$, 50.1 to 50.3 kJ mol$^{-1}$, and 51.8 to 52.6 kJ mol$^{-1}$ in the pre-warming

SOC-varied simulations for CON, and the posterior means of $Ea_V$, $Ea_{VU}$, $Ea_K$, and $Ea_{KU}$ respectively ranging from 58.5 to 74.8 kJ mol$^{-1}$, 50.2 to 51.1 kJ mol$^{-1}$, 25.8 to 42.4 kJ mol$^{-1}$, and 49.0 to 49.8 kJ mol$^{-1}$ for AWB.
However, these values are still within the ranges of organic matter decomposition activation energies, which have been empirically estimated to exceed 100 kJ mol$^{-1}$ at their highest in the A-horizons of temperate soils (Steinweg et
al., 2013), suggesting that the $Ea$ posterior means, aided by prior truncation, effectively remained within biologically realistic space across all observed pre-warming C densities. The presence of higher $Ea_S$ posterior means
also agreed with the empirical trends of higher activation energies for the degradation of SOC-related organic compounds and lower activation energies for the degradation of material associated with microorganisms.

We found it less useful to compare the posterior means of other fitted parameters including the C pool transfer coefficients, C use efficiency $E_C$, and $V_{max}$ to empirical estimates for biological benchmarking purposes.

Unitless parameters like transfer coefficients and $E_C$ defy straightforward interpretation, measurement, and estimation from experiments (Bradford and Crowther, 2013). Very different values can be found based on whether
substrate-specific or substrate-nonspecific assumptions and methods are used (Geyer et al., 2019; Hagerty et al., 2018). $V_{max}$ parameters are not unitless but display even higher variance than the bounded C transfer and efficiency
coefficients. The $V_{max}$ parameter corresponding to a specific enzyme can vary over orders of magnitude when the sensitivity of the enzyme to an interval of temperatures is considered (Nottingham et al., 2016). The process of
consolidating experimental substrate-specific and substrate-nonspecific measurements into a single number to correspond to a model $V_{max}$ value introduces further complications and uncertainty, rendering comparisons of
potentially drastically different $V_{max}$ values less informative regarding model biological realism.

### 4.3 HMC Parameter Space Exploration

Truncating prior and posterior parameter distributions proved useful for establishing biological constraints and only modestly deformed posterior densities for AWB and CON. From SOC100 to SOC200, CON and AWB

posterior densities showed little or no deformation from typical normal distribution shapes. Moderate posterior density deformation was observed for some parameters in both models at SOC50 and SOC75, namely $Ea_S$ for CON
and $E_{C_{ref}}$ for AWB (Supplemental Fig. 11). Even so, most of the other parameter posterior densities still remained undeformed at those SOC values. Thus, prior truncation generally did not prevent posterior means from falling
within biologically realistic intervals, suggesting that priors were appropriately informed and chosen.

        A small frequency of divergent transitions was detected in the AWB HMC simulations. Divergent

transitions can be thought of as algorithm trajectory errors arising during the HMC's exploration of a convoluted region of parameter space; a more thorough description of the theory, computation, and implications of divergent
transitions can be found in literature focusing on the Hamiltonian Monte Carlo algorithm (Betancourt, 2016, 2017). The number of divergent transitions generally increased as the pre-warming MIC-to-SOC steady state ratio was
reduced (Supplemental Fig. 9). Prior truncation and the fixing of select parameters to constrain the pre-warming steady state mass values for biological realism could have played a combined role in generating the Markov chain
divergences by hindering the smooth exploration of parameter space. We were unable to eliminate divergent transitions by adjusting HMC parameter proposal step size, suggesting that other methods, such as modification of
the HMC algorithm itself or introduction of auxiliary parameters to AWB that reduce correlation between existing model parameters may be more applicable in reducing divergent transitions in our case (Betancourt and Girolami,
2015). Additionally, the interaction between the ranges of values used for the prior distributions and the limited number of observations in the data set could have contributed to the shaping of geometric inefficiencies (Betancourt,

2017).

        It is possible that the instability that prevented consistent solving and HMC exploration of AWB under

SOC50 could be traced to the forward Michaelis-Menten formulation of decomposition and uptake kinetics used in the present version of the AWB model (Supplemental Appendix 1 Equations A7, A8). We initialized the system
with a small DOC density lower than that of MIC at 0.1 mg C g$^{-1}$ soil. Since DOC was in the denominator of these decomposition and uptake expressions, those expressions could become larger than tolerable for the system in
certain parameter regimes.

        Some suggestions for the re-parameterization of AWB to improve model stability have been proposed that

could reduce or even eliminate divergent transitions by facilitating a smoother and steadier parameter space
conducive for HMC exploration. One intermediate possibility would be to modify AWB to use reverse Michaelis-Menten kinetics, which would replace the DOC term in the denominators of the decomposition and uptake expressions with the larger MIC term. The use of reverse instead of Michaelis-Menten dynamics has been used to
stabilize and constrain other SBMs (Sulman et al., 2014; Wieder et al., 2015b). A more extensive re-formulation involves the replacement of Michaelis-Menten expressions with equilibrium chemistry approximation (ECA)
kinetics, which would increase the number of denominator terms in decomposition expressions for further stability. ECA equations have been shown to be more consistent in behavior and robust to parameter regime variation than
their Michaelis-Menten counterparts, and thus have been encouraged as a wholesale replacement for Michaelis-Menten formulations (Tang, 2015; Wang and Allison, 2019). These re-parameterizations should be implemented
and examined in future work that involves sampling and computation of AWB posteriors.

### 4.4 Outlook and Conclusions

Recent SBM comparisons have been unable to demonstrate the superiority of one model over another because the uncertainty boundaries of the data were not sufficient for distinguishing model outcomes (Sulman et al.,
2018; Wieder et al., 2014, 2015b, 2018). Similar to these previous studies, our results indicate that more data is needed to constrain and differentiate between model posterior predictive distributions. Conditional on the meta-
analysis data set, CON demonstrates superior quantitative goodness-of-fit over AWB, but we are not confident that the relative model parsimony of CON and other linear first-order models makes them universally more suitable for
predictive use.

Consequently, future SBM comparisons would benefit from additional data collection efforts sourced from

long-term ecological research experiments to globally verify the strengths and limitations of linear versus non-linear SBMs, including CON and AWB, in Earth system modeling. The limited number of longitudinal soil warming
studies presents a challenge for facilitating site-specific model comparisons. We addressed this issue by using meta-analysis data to aggregate warming responses across sites, but this approach does not provide site-specific
parameters. Additional data from ongoing and future field warming studies in the vein of the Harvard Forest and Tropical Responses to Altered Climate experiments that demonstrate more varied flux dynamics over time than the
meta-analysis data set will be of critical importance for model testing (Melillo et al., 2017; Wood et al., 2019). Model parameters could also be better constrained through the use of multivariate data sets, for example microbial
biomass dynamics in addition to soil respiration.

Our approach can be expanded to compare the predictive accuracies of linear microbial-implicit models to

those of recently developed non-linear microbial-explicit SBMs that are much larger than AWB, such as CORPSE (Sulman et al., 2014) and MIMICS (Wieder et al., 2014). Such comparisons will help broadly determine if inclusion
of more detailed microbial dynamics in models offers predictive advantages that can overcome the overfitting burdens associated with an increase in parameter count. With the appropriate data sets, our approach can also be
applied to consider the predictive performance of SBMs that describe the cycling of nitrogen (N), phosphorus (P), and other limiting nutrients in addition to C dynamics. Models that represent N and P mineralization have yet to see
extensive head-to-head statistical benchmarking against C-only models with respect to predictive use (Manzoni and Porporato, 2009). With models growing ever larger in size and specificity, there is a need to verify whether detailed
representation of microbial processes and the cycling of limiting nutrients are worth the increase in variable, parameter, and equation counts. After all, "the tendency of more recent models towards more sophisticated (and
generally more mathematically complex) approaches is not always paralleled by improved model performance or ability to interpret observed patterns" (Manzoni and Porporato, 2009).
The data assimilation and posterior sampling of complex models in future work comes with computing performance challenges. Markov chain Monte Carlo algorithms are effective for exploring multidimensional
parameter space but are limited by temporal and computational expense, particularly when it comes to fitting non-linear differential equation models (Calderhead et al., 2009; Nemeth and Fearnhead, 2019). Time per Markov chain
iteration drastically increases with number of parameters and data points. In fact, the present speed limitations of the family of HMC algorithms make it necessary to use a hybrid approach utilizing Monte Carlo and deep learning
algorithms for parameter estimation at a global scale; Monte Carlo fitting is used to constrain parameter estimates at a site-based scale before those estimates are tuned globally by deep learning using spatial information derived from
satellite maps (Tao et al., 2020). However, Monte Carlo algorithms are still the optimal methods for posterior computation (Duan et al., 2018) and are necessary for Bayesian model comparisons conditional on site-based data.
Consequently, recent Monte Carlo algorithm innovations and developments that offer theoretical speed improvements by trading thorough posterior sampling for numerical efficiency have been encouraging and are ripe
to be tested in future SBM comparisons involving more complex models and larger data sets. These developments

include stochastic gradient Monte Carlo sampling methods, a class of techniques in which a posterior is approximated by fitting to a small subset of data at each iteration rather than estimated through exhaustive sampling (Ma et al., 2015), and Gaussian process acceleration, in which a smooth distribution of likely solutions for a differential equation system is specified and sampled in place of explicitly solving for the state variables during every Markov chain iteration (Dondelinger et al., 2013; Wang and Barber, 2014).

Alongside advances in Monte Carlo algorithms, additional developments in Bayesian cross-validation and information criteria measures are also available for practical trialing in soil biogeochemical data assimilation. Gelman et al. have proposed a stable Bayesian counterpart of frequentist $R^2$ defined as "the variance of the predicted values divided by the variance of predicted values plus the expected variance of the errors" that allows for more intuitive and direct comparison to $R^2$ (Gelman et al., 2019). A Bayesian $R^2$ distribution provides a signal about the absolute rather than relative goodness-of-fit of an associated posterior predictive distribution to the data. Bürkner et al. (2019) have proposed a leave-future-out (LFO) cross-validation metric which is formulated to estimate relative model predictive accuracy for hypothetical time series data occurring after existing experiment observations. LFO and LOO are computed similarly, and LOO can also be used for time series data, as we demonstrated in this study. However, the algorithmic differences between LFO and LOO make them better suited for different goals. LOO does not inform about the quality of model fits for hypothetical samples collected after final reported measurements and is more appropriate for estimating out-of-sample model predictive accuracy for hypothetical data samples taken between the interval of observed measurement times (Vehtari et al., 2017).

The development of our formalized, statistically rigorous approach for model comparison and evaluation is a critical step toward the goal of projecting global SOC levels and soil emissions throughout the 21$^{st}$ century. Our initial results indicate promise in continued refinement and expansion of our approach to evaluate the predictive performance of linear and non-linear SBMs. The future integration of updated Markov chain algorithms and Bayesian predictive accuracy metrics into our framework will expand the ability to efficiently and thoroughly compare differential equation models, even if they vary widely in structure and complexity.

**Code and Data Availability**

The R scripts, Stan code, and respiration data set used for HMC model fitting along with the original soil respiration meta-analysis data set (Romero-Olivares et al., 2017) are available from the directory located at https://osf.io/7mey8/?view_only=af1d54f858c34c41ab4854551d015896 (Xie et al., 2020).

**Author contribution**

SDA and HWX designed the study with assistance from MG. HWX and ALR performed the data cleaning and analysis. HWX wrote the necessary code for the study with assistance from SDA. SDA and HWX prepared the paper with suggestions from MG.

**Competing interests**

The authors declare they have no conflict of interest.

**Acknowledgments**

We would like to thank Stan development team members Aki Vehtari (Aalto University), Michael Betancourt, Bob Carpenter (Flatiron Institute), Ben Bales (Columbia University), Charles Margossian (Columbia University), and Sebastian Weber (Novartis) for their patient help with Stan code implementation and troubleshooting. We would also like to thank both anonymous reviewers for their valuable and constructive comments, which not only aided in the revision of the manuscript but also provided valuable insights to guide future work.

**Financial support**

This research was supported by funds from the National Science Foundation under grant DEB-1900885, the U.S. Department of Energy Office of Science BER-TES program under grant DESC0014374, and the National Institutes of Health T32 Training Program under grant EB009418.

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

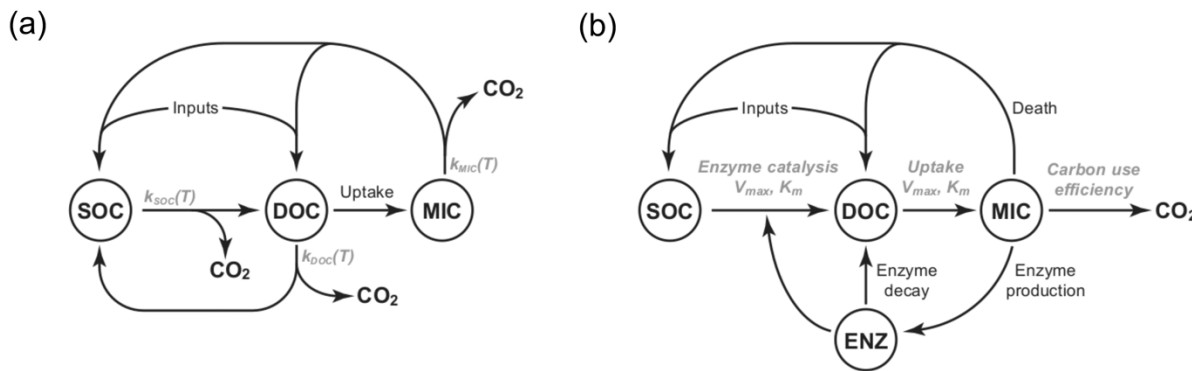

**Figure 1:** Diagrams of the pool structures of the **(a)** CON model; and **(b)** AWB model drawn from Allison et al.,
(2010). Pools are shown within circles including soil organic carbon (SOC), dissolved organic carbon (DOC), and
microbial (MIC) pools. AWB has SOC, DOC, and MIC pools as in CON, but also an extra enzymatic (ENZ) pool.
AWB additionally differs from CON in its non-linear feedbacks and assumption that MIC can influence SOC-to-
DOC turnover through the ENZ pool.





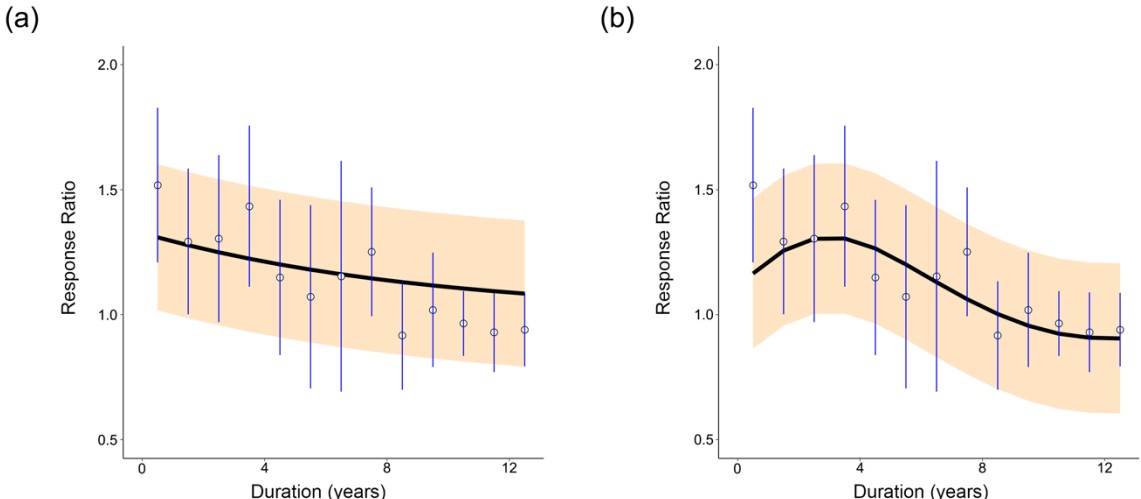

**Figure 2:** Distribution of fits of **(a)** CON; and **(b)** AWB to the meta-analysis data from Romero-Olivares et al., (2017). Open circles show the meta-analysis data points. Blue vertical lines mark the 95% confidence interval for each data point calculated from the pooled standard deviation. The black line indicates the mean model response ratio fit. The orange shading marks the 95% posterior predictive interval for the fit. For **(a)**, pre-warming steady state soil C densities were set at SOC = 100 mg C g$^{-1}$ soil, MIC = 2 mg C g$^{-1}$ soil, DOC = 0.2 mg C g$^{-1}$ soil. For **(b)**, pre-warming steady state soil C densities were set at SOC = 100 mg C g$^{-1}$ soil, MIC = 2 mg C g$^{-1}$ soil, DOC = 0.2 mg C g$^{-1}$ soil, and ENZ = 0.1 mg C g$^{-1}$ soil.

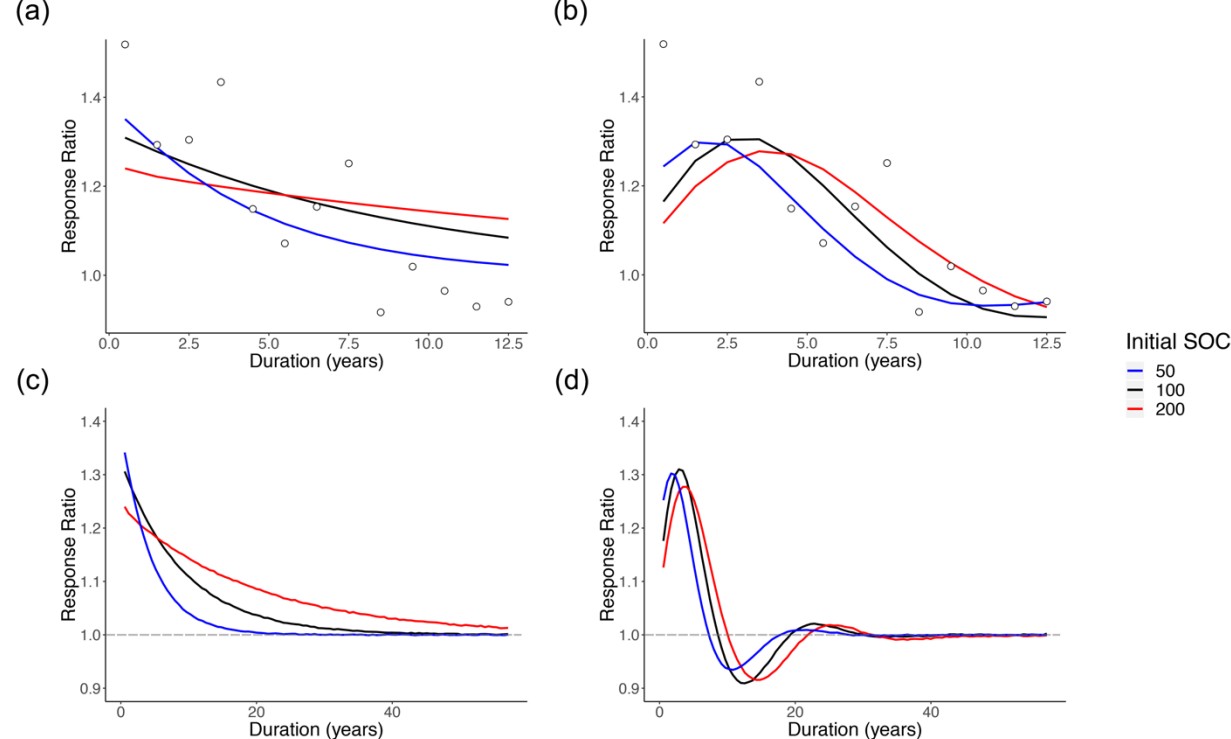

**Figure 3:** Intra-model comparisons of mean posterior predictive response ratio fits for AWB and CON across different MIC-to-SOC ratios. Open circles show the meta-analysis data points for reference. The blue, black, and red
lines indicate model mean fits corresponding to different pre-warming-perturbation steady state SOC values of 50 mg C g$^{-1}$ soil, 100 mg C g$^{-1}$ soil, and 200 mg C g$^{-1}$ soil. The dashed gray line indicates the steady state expectation at
the response ratio of 1.0. Mean fits are plotted in order of **(a)** CON; and **(b)** AWB over the time span of the data and **(c)** CON; and **(d)** AWB over 57 years.

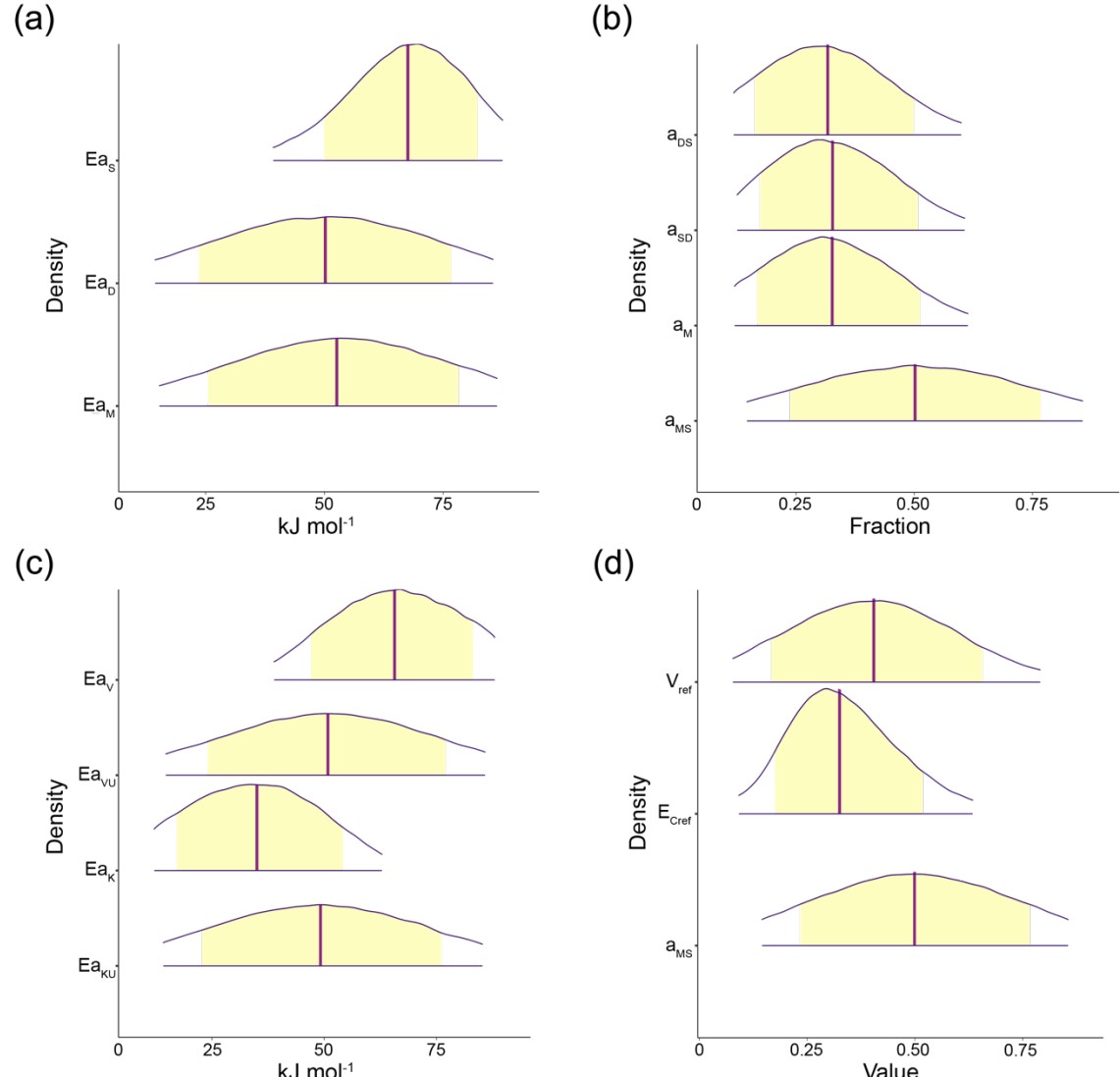

**Figure 4:** 95% probability density credible areas for model parameters corresponding to pre-warming steady state SOC = 100 mg C g$^{-1}$ soil, DOC = 0.2 mg C g$^{-1}$ soil, MIC = 2 mg C g$^{-1}$ soil, and (for AWB) ENZ = 0.1 mg C g$^{-1}$ soil. Yellow shaded regions represent 80% credible areas and vertical purple lines indicate distribution mean. **(a)** CON activation energy parameters $Ea_S$, $Ea_D$, and $Ea_M$; **(b)** CON C pool partition fraction parameters $a_{DS}$, $a_{SD}$, $a_M$, and $a_{MS}$; **(c)** AWB activation energy parameters $Ea_V$, $Ea_{VU}$, $Ea_K$, and $Ea_{KU}$; **(d)** AWB parameters $V_{ref}$, $E_{C_{ref}}$, and $a_{MS}$. $V_{ref}$ is the SOC V$_{max}$ at the reference temperature 283.15 K, $E_{C_{ref}}$ is the carbon use efficiency fraction at the reference temperature, and like its CON counterpart, the AWB $a_{MS}$ parameter is the fraction parameter representing the proportion of dead microbial biomass C transferred to the SOC pool. Parameter units are displayed in Supplemental Table 1. Credible areas for AWB parameters $V_{U_{ref}}$ and $m_t$ are shown in Supplemental Fig. 2 because of differing horizontal axes scales.

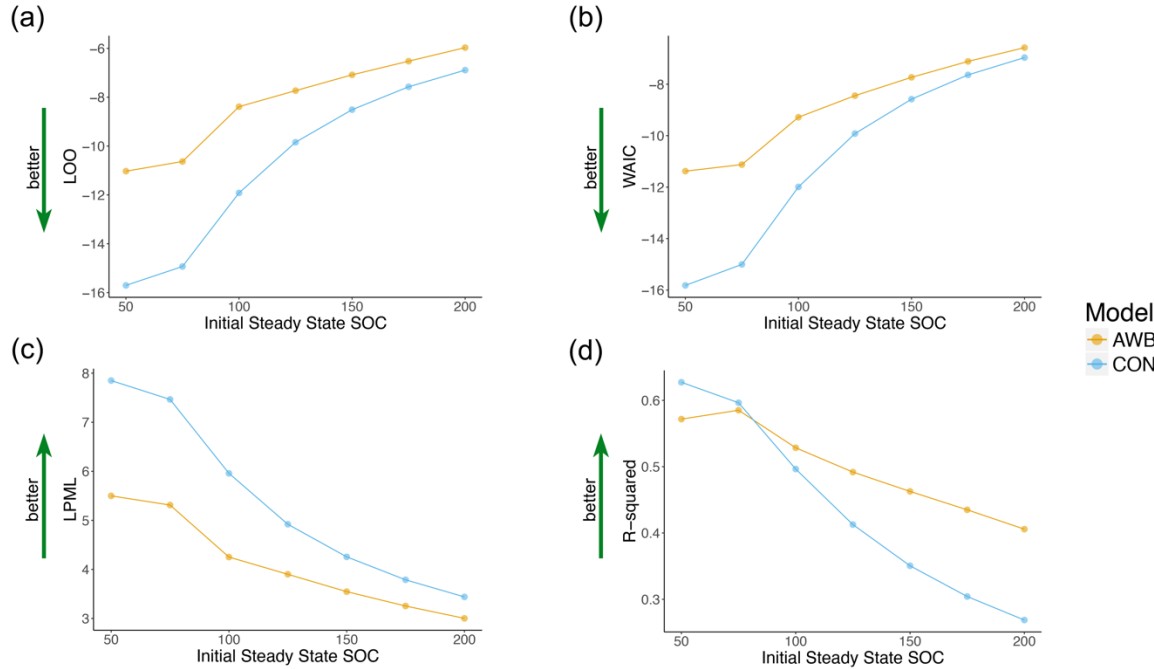

**Figure 5:** Goodness-of-fit metrics plotted against initial steady state SOC for AWB and CON models for **(a)** LOO; **(b)** WAIC cross-validation; **(c)** LPML; and **(d)** $R^2$ values. Pre-perturbation steady state MIC, DOC, and ENZ (for AWB) is held constant as pre-perturbation SOC is varied.