# Peer review of "A Bayesian Approach to Evaluation of Soil Biogeochemical"

_Biogeosciences, 2020_

## Referee Comment (RC1) · Anonymous Referee #1 · 17 Mar 2020

General comments: The manuscript "A Bayesian Approach to Evaluation of Soil Bio-geochemical Models" by Hua W. Xie et al. presents a Bayesian approach to soil biogeo-chemical models. This study provides valuable insight – soil biogeochemical models need to be assessed by Bayesian goodness-of-fit metrics, not a widely used metric, i.e., R-squared. Furthermore, they compared between linear and nonlinear models – first-order linear ordinary differential equation and non-linear Michaelis-Menten func-tion, respectively. Despite somewhat expected the main conclusion, e.g., more data could help model to constrain parameters so that it could be possible to define the strengths and limitations of linear vs. nonlinear models; this comparison based on the Bayesian perspective suggested that soil biogeochemical model(s) need to consider the matric of the Bayesian goodness-of-fit for a better model selection, having strong

predictive skills.

This manuscript fits the scope of the Biogeosciences journal and identifies the potential implement for current generation of soil biogeochemical models. While the approach and analysis presented in the manuscript are generally correctly conducted and concluded, there are some minor issues in the manuscript. I offer specific comments for improving the manuscript below.

Specific comments:

Methods

L 86-91: lacks model descriptions; and nitrogen-related increases in complexity has not been addressed in the entire manuscript. I suggest you may discuss in the discussion. Also, figure 1 has not been mentioned in the manuscript

L 166: Log Pseudomarginal Likelihood (LPML) has popped up without prior introduction

Results

L 185: "The difference in curve shape (Fig. 3a, b)..."

L 189: Is it different between "95% confidence interval" and "95% model response ratio credible interval"?

L 196: a bit confused as well as missing figure annotation. It would be better to choose clear points to address why CON and AWB are showing differences

L 198: rewording to emphasize "how the steady state pool size ratio has been changed based on increasing MIC"; the unit should be mg C g-1(uppercase); Please check other lines as well

L 199: need to clarify. By the way, what is the function of the trend lines? Have you tried polynomial function? It seems similar patterns between them.

L 203-206: Is it possible to replace the supplemental figure 3 to represent SOC loss rather than "SOC fraction remaining"? It is difficult to interpret.

L 205: "decreased from 16.3 to 11.3 %". Please check other lines as well

L 218: R2; Annotation for varying SOC: "SOC = 50 -> SOC50"

Discussion

L 231: not sure this manuscript compared models through AIC and DIC with WAIC and LOO

L 255-259: which Figures showing this? Also, the sentences are too complex. Please re-write simpler sentences

L 273: is 50 mg SOC g-1 soil same in line 218 (SOC =50)? Please use consistent unit

L 275: Supplemental Table 3?

L 316: Supplemental Fig. 5

L 322: Supplemental Fig. 8a, b...

Typing error

1. Please check upper case expression; r-squared is R2 (uppercase)

2. L 156: mg C g-1 soil

3. Put period after abbreviation of figure, e.g., Fig. xx

4. Please double-check figures and table numbers

5. Supplemental or supplementary?

---

## Referee Comment (RC2) · Anonymous Referee #2 · 20 Mar 2020

This study fits two soil decomposition models (one linear and one nonlinear) to response ratios of CO2 fluxes to warming collected from field experiments. They estimate full parameter distributions using MCMC in a Bayesian framework. This is a great way to fit and evaluate models, as well as to place uncertainty bounds on subsequent predictions.

I wanted this paper to consider more the implications of its findings. For example I would have expected the introduction to focus more on why temperature response is especially important since that is the dataset that the authors focus on here. I was also interested to compare the performance of the two models. The performance metrics were presented but their results are never discussed. Why was it that CON generally performed better at most SOC densities? What are the estimated parameter ranges

compared to other models/literature values? For example the activation energy for SOM decomposition seems to be higher than the activation energy for other processes like uptake which may have some interesting implications. This is partially an interpretation of the scope of Biogeosciences vs. say Geoscientific Model Development, but I expected a little more interpretation of the processes underlying the performance metrics.

L62-66: Citation for this discussion of R2?

Also there are other metrics for evaluating Bayesian models that are not discussed here – you don't need an exhaustive review but ROC/AUC and BIC seem common.

L125: 0.9995 and 0.001 seems like extreme adaptation and step sizes to me, causing the model to take many small steps. If this is a supported strategy, can you provide a citation or justify further?

L191: As written, implies that AWB performs better because it has a higher RR in subsequent years after the first year, but the data show that the first year has the highest RR, so CON seems to correspond more closely to this. In the discussion, you can bring up the potential realism of oscillations given the Harvard Forest long term warming experiment.

L325-333: This discussion of R2 and other cost metrics seems repetitive to the introduction.

It seems like the performance metrics would be yet better with a lower SOC density (<50 mg SOC/g soil), if it were possible to achieve them without the AWB instability. I think you could fix the instability by changing your decomposition/uptake kinetics. Right now in the uptake equation DOC is in the denominator but its initial concentration is much smaller than MIC. So you can either flip to Reverse M-M for uptake or use ECA where both quantities (SOC and ENZ or MIC and DOC) are in the denominator <- this may be harder to fit because it will be more constrained, but it is also harder to break.

---

## Author Comment (AC1) · 28 Mar 2020

We are grateful to the reviewer for the time and effort he, she, or they has spent on their feedback and suggestions. The comments are insightful and helpful. We address specific reviewer comments below.

> L 86-91: lacks model descriptions; and nitrogen-related increases in complexity has not been addressed in the entire manuscript. I suggest you may discuss in the discussion. Also, figure 1 has not been mentioned in the manuscript

We will describe the model structures in greater detail in the methods section of the paper and add a reference to Figure 1. Models representing N-cycling in addition to C-cycling will be addressed in a revision of the discussion section. We will also consider future Bayesian model comparisons involving N-cycling and other more complex soil biogeochemical models in a new discussion paragraph. Potential challenges arising in future model comparisons of more complex models, such as the difficulty of solving for steady states in some non-linear systems and increased HMC simulation time, will be detailed.

> L 166: Log Pseudomarginal Likelihood (LPML) has popped up without prior introduction

Log Pseudomarginal Likelihood (LPML) is a cross-validation metric that is calculated similarly to Leave-one-out (LOO). As described in the manuscript, LPML does not calculate effective parameter count, which was introduced with LOO. Hence, it was included in the study to serve as a comparison to LOO and underscore the goodness-of-fit calculation without penalizing for overfitting. To make the introduction of LPML less jarring, we will mention it in the paragraph located in the introduction section that provides background on LOO and WAIC.

> L 185: The difference in curve shape (Fig. 3a, b)

We will fix this figure annotation and related ones referring to more than one figure pane.

> L 189: Is it different between 95% confidence interval and 95% model response ratio credible interval?

"Credible interval" should actually be changed to say "posterior predictive interval." That will be fixed in the revision. The 95% confidence interval and 95% posterior predictive interval are indeed different. In this case, the confidence interval is a frequentist measure corresponding to estimates of the true value of the flux data measurement, while the Bayesian posterior predictive interval indicates the range of model predictions for mean flux response ratio.

> L 196: a bit confused as well as missing figure annotation. It would be better to choose clear points to address why CON and AWB are showing differences

We will revise this paragraph to include the additional appropriate figure citations. The paragraph will be re-structured and re-worded to compare model outputs and data at specific time points.

L 198: rewording to emphasize how the steady state pool size ratio has been changed-based on increasing MIC; the unit should be mg C g-1(uppercase); Please check other lines as well

We will re-word more simply to emphasize that the pre-warming SOC-to-MIC ratio was changed by increasing MIC while holding SOC steady. $g^{-1}$ and other inverse units show up appropriately as exponents in the Microsoft Word .doc file, but regrettably do not appear as exponents following the *Biogeosciences Discuss* PDF conversion. We will try uploading the file with some other steps to try to fix the exponent issue with our subsequent revision.

L 199: need to clarify. By the way, what is the function of the trend lines? Have you tried polynomial function? It seems similar patterns between them.

The intent of the posterior predictive mean fit trend lines in Supp. Fig. 1 was to show the moderate, but consistent effect of increasing pre-warming MIC on the AWB model output slopes. In our revision, we can emphasize that the influence of pre-warming MIC on slope magnitude is limited to the AWB model. My writing was not clear for this paragraph. We can also edit the paragraph to remove reporting of the influence of MIC on AWB slope and streamline this section, as this result is not central to the manuscript.

By polynomial function, I assume you mean the polynomial analytic function

$$f(x) = \sum_{n=0}^{\infty} c_n (x)^n = c_0 + c_1 x + c_2 x^2 + c_3 x^3 + ...$$

We could exactly fit the data using a polynomial analytic function with sufficiently many $c_n x^n$ terms, but the goal of the manuscript was not to find the best-fitting arbitrary model. Instead, we sought to demonstrate the feasibility of rigorous Bayesian model comparisons for more complex dynamical ODE systems with the hope that further conclusions regarding model mechanisms and structure could be made following future model comparison results. Fitting analytic polynomial model would not be able to contribute feedback towards the refinement or rejection of elements of dynamical model structure or formulation. Additionally, the polynomial model would be prone to overfitting and effective parameter count penalties from the LOO and WAIC computation.

L 203-206: Is it possible to replace the supplemental figure 3 to represent SOC loss rather than SOC fraction remaining? It is difficult to interpret.

We wanted the y-axis of Sup. Fig. 3 to align with Sup. Fig. 10 to show that the SOC remaining in our models was in the range of change of SOC observed in soil warming experiments across various soil types. This indicates that our models were within the realm of biological realism. However, we can see that the labeling would make Sup. Fig. 3 confusing. We do feel that changing Sup. Fig. 3 to show an absolute level of SOC loss in terms of density would make it less useful for direct comparison to Sup. Fig. 10. The fraction change is what we want to observe; soils vary dramatically in SOC concentration and the fraction of change in SOC provides more information about the biological realism of the model following preturbation from initial conditions than the change in absolute SOC density.

Consequently, we propose the following changes to Sup. Fig. 3 and Sup. Fig. 10:

- We will re-arrange and re-number the figures in the supplement so that Sup. Fig. 3 and Sup. Fig. 10 follow each other to reduce confusion.

- We will change the vertical limits and add a dashed horizontal line at 1.0 in Sup. Fig. 3 and Sup. Fig. 10 to reflect the divide between SOC gain and loss after warming.

- We will add vertical arrows going down from the horizontal line in Sup. Fig. 3 labeled with "SOC loss" to clarify the direction of greater SOC losses.

(a)

[Figure]

(b)

[Figure]

[Figure]

Updated Supp. Fig. 10. To be re-numbered.

[Figure]

L 205: decreased from 16.3 to 11.3 %. Please check other lines as well

We will fix occurrences of redundant symbols in the revision.

L 218: R2; Annotation for varying SOC: "SOC = 50 -> SOC50"

We will look into and fix the aforementioned exponent rendering issues and standardize the notation for pre-warming steady state densities. We commend this effective notation suggestion.

L 231: not sure this manuscript compared models through AIC and DIC with WAIC and LOO

Yes, AIC and DIC were not computed and compared for reasons of redundancy, stability, and algorithmic limitations. AIC relies on a maximum likelihood estimate which cannot be calculated from non-uniform priors in a Bayesian setting [2]. AIC is more readily deployed in frequentist model comparisons in which normally distributed prior information is not used. DIC can be computed under Bayesian settings, but is an approximation of WAIC that is calculated from posterior means, whereas WAIC involves integration over the posterior distribution sample [1]. Since WAIC can already be readily calculated, has been demonstrated to be more stable and accurate than DIC in much statistics literature, and is itself an approximation of LOO [2], we felt that calculating and plotting WAIC, LPML, and LOO was sufficient for illustrating the influence of the pre-warming steady state ratios on goodness-of-fit. We initially felt that some of this information was too technical and distracting to discuss in a paper not being submitted to a statistics journal, but we now agree that we should summarize this information with citations in the introduction or methods section to justify our use of WAIC, LPML, and LOO.

L 255-259: which Figures showing this? Also, the sentences are too complex. Please re-write simpler sentences

There was an oversight here, so we will add a figure citation in this paragraph referring to Supp. Fig. 3a, b which supports the assertions made in that paragraph. The sentences will be rewritten to be less convoluted.

L 273: is 50 mg SOC g-1 soil same in line 218 (SOC =50)? Please use consistent unit

The line here was not referring to the specific SOC50 simulation and HMC run, but to the observation that we were unable to initialize simulations with pre-warming SOC below 50 mg C g$^{-1}$, so we felt that it was appropriate to write the full units out rather than abbreviate here.

L 275: Supplemental Table 3?

Will be corrected.

L 316: Supplemental Fig. 5

Will be corrected.

L 322: Supplemental Fig. 8a, b

Will be corrected.

Typing error

1. Please check upper case expression; r-squared is R2 (uppercase)
2. L 156: mg C g-1 soil
3. Put period after abbreviation of figure, e.g., Fig. xx
4. Please double-check figures and table numbers
5. Supplemental or supplementary?

We will fix upper case, exponent, abbreviation, and numbering issues. We will replace isntances of "supplementary" with "supplemental." We appreciate the close reading, fixes, and suggestions, and will address them in our revision.

**References**

[1] M. Betancourt. A unified treatment of predictive model comparison, 2015. arXiv:1506.02273.

[2] A. Gelman, J. Hwang, and A. Vehtari. Understanding predictive information criteria for Bayesian models. *Statistics and Computing*, 24(6):997–1016, 2014.

---

## Author Comment (AC2) · 28 Mar 2020

We are grateful to the reviewer for the thoughtful and constructive feedback. We address the specific reviewer comments below.

> Why was it that CON generally performed better at most SOC densities? What are the estimated parameter ranges compared to other models/literature values? For example the activation energy for SOM decomposition seems to be higher than the activation energy for other processes like uptake which may have some interesting implications.

We feel that the chief reason that CON performed relatively better than AWB at most SOC densities by information criteria and cross-validation was because of overfitting and parameter count. By R2, AWB could fit the data slightly better by absolute residual sums than CON at most pre-warming SOC (Fig. 5), but AWB has more parameters and was penalized for that, and we were reluctant to over-reach on conclusions. However, you make an excellent point about the need for more discussion about the biological implications of the parameter posteriors and fitting results. Discussion of the biological realism of the models we used is more limited to the extent of SOC loss over time in the current iteration of the manuscript. Thus, in our revision, we will discuss more how our posteriors from each model compare with empirical results from literature. We will also add our explanation for why we think the mean posterior SOM decomposition activation energy ended up higher than the mean posterior activation energies of other processes. We believe that was the case because SOC decomposition is generally the the rate-limiting step in C-cycling systems that represent microbial activity. If SOM decomposition Ea were too low, the soil C would cycle too fast and result in a poorer fit to the data set.

> L62-66: Citation for this discussion of R2? Also there are other metrics for evaluating Bayesian models that are not discussed here you dont need an exhaustive review but ROC/AUC and BIC seem common.

For the discussion of R2, we will cite Kvålseth 1985[3], Spiess and Neumeyer 2010[6], and Gelman et al. 2019[1].

We initially did not discuss BIC due to its similarity to AIC. BIC is closely related to AIC and its computation is dependent on the pointwise maximum likelihood estimate from frequentist methodology[2]. Hence, BIC, contrary to its name, is not a fully Bayesian metric calculated from the posterior distribution. However, as BIC is indeed used to compare out-of-sample predictive accuracy of groups of models, we agree that it should mentioned and will revise our manuscript to include BIC in the introduction.

We would like to avoid discussing ROC/AUC because they are indicators of the prediction accuracy of binary classifier models trained and tested on categorical data[5]. In our case, we were specifically looking at metrics that estimated out-of-sample prediction accuracy or goodness-of-fit for models conditional on ordinal data with elements in $\mathbb{R}$. LOO/WAIC and ROC/AUC correspond to fundamentally different model and data types, so we feel that ROC/AUC would be off-topic.

> L125: 0.9995 and 0.001 seems like extreme adaptation and step sizes to me, causing the model to take many small steps. If this is a supported strategy, can you provide a citation or justify further?

To start, it is worth clarifying the differences between traditional MCMC and the Hamiltonian Monte Carlo algorithm that Stan uses. The HMC is not a random walk algorithm, and each of its trajectories are deterministically calculated via Hamiltonian dynamics. An MCMC step size parameter is fixed through the duration of the sampling. In contrast, the step size is tuned at each step based on the adapt delta and calculated Hamiltonian trajectory. I should have clarified in the manuscript (and will do so in the revision) that the step size used was the initial step size. Consequently, starting with a small step size does not mean that the HMC algorithm takes fixed steps of the same size for the rest of the chain (refer to this page from the Stan documentation for more detail).

The adapt delta and initial step sizes were set as such in an effort to reduce the number of divergent transitions during the HMC sampling, which was an issue for the AWB model. As can be seen in the supplement, divergent transitions were still detected for AWB following the implementation of the strategy. This indicates the future need to re-parameterize and re-formulate AWB to obtain smoother and more stable parameter space geometries to be explored. However, the divergences occurred at a reduced rate compared to using Stan's default adapt delta of 0.8, so the strategy proved helpful for obtaining a suitable number of posterior samples.

We did not encounter any divergent transitions with CON, but mirrored the HMC parameters for both models since the HMC parameters ultimately do not alter the overall exploration of the parameter space. Increasing adapt delta results in less sensitive tuning of the step size per iteration. In our case, this does mean our step size will be smaller on average as our initial step size will be less responsive to tuning and slower to increase, but the algorithm will be better at navigating geometrically trickier parameter space to generate fewer trajectory divergences. This trade-off comes at the cost of more computationally expensive and less efficient calculated Hamiltonian trajectories. The smaller steps correspond to a drive for more changes in trajectory direction to cover the same amount of ground. With sufficient chain length and samples (which our Bayesian diagnostics indicates that we obtained), the parameter space should still have been adequately explored [4].

The strategy of using higher adapt delta and lower initial step sizes is less documented in formal literature, but has been previously used by other Stan users to mitigate divergent transitions (see this link and this link for further discussion). It was important for us to maximize the amount of samples we obtained for AWB, so the increased computational time per iteration was a worthwhile tradeoff for us.

> L191: As written, implies that AWB performs better because it has a higher RR in subsequent years after the first year, but the data show that the first year has the highest RR, so CON seems to correspond more closely to this. In the discussion, you can bring up the potential realism of oscillations given the Harvard Forest long term warming experiment.

We will do this.

> L325-333: This discussion of R2 and other cost metrics seems repetitive to the introduction.

We will prune redundant information in this part of the discussion.

It seems like the performance metrics would be yet better with a lower SOC density (¡50 mg SOC/g soil), if it were possible to achieve them without the AWB instability. I think you could fix the instability by changing your decomposition/uptake kinetics. Right now in the uptake equation DOC is in the denominator but its initial concentration is much smaller than MIC. So you can either flip to Reverse M-M for uptake or use ECA where both quantities (SOC and ENZ or MIC and DOC) are in the denominator <- this may be harder to fit because it will be more constrained, but it is also harder to break.

These are perceptive and insightful modeling suggestions that we greatly appreciate. In the discussion section of this manuscript, we will add some sentences to describe the importance of exploring the effect of changes to microbial-explicit model structures including the ones you proposed on data fitting and posterior sampling in subsequent work. Then, we will apply those AWB model modifications to an in-progress follow-up model comparison project that fits models to a different, larger data set from Harvard Forest. We feel that these reparameterization suggestions could help reduce or eliminate the number of divergent transitions generated during AWB HMC posterior sampling.

**References**

[1] A. Gelman, B. Goodrich, J. Gabry, and A. Vehtari. R-squared for Bayesian Regression Models. *The American Statistician*, 73(3):307–309, jul 2019.

[2] A. Gelman, J. Hwang, and A. Vehtari. Understanding predictive information criteria for Bayesian models. *Statistics and Computing*, 24, jul 2013.

[3] T. O. Kvålseth. Cautionary Note about R2. *The American Statistician*, 39(4):279–285, nov 1985.

[4] S. Livingstone, M. Betancourt, S. Byrne, and M. Girolami. On the Geometric Ergodicity of Hamiltonian Monte Carlo, Jan 2016. arXiv:1601.08057.

[5] T. Saito and M. Rehmsmeier. The precision-recall plot is more informative than the ROC plot when evaluating binary classifiers on imbalanced datasets. *PloS one*, 10(3):e0118432–e0118432, mar 2015.

[6] A. N. Spiess and N. Neumeyer. An evaluation of R2 as an inadequate measure for nonlinear models in pharmacological and biochemical research: A Monte Carlo approach. *BMC Pharmacology*, 10(1):6, dec 2010.